# A molecular T-pentomino for separating BTEX hydrocarbons

Christopher J. Hartwick[1], Eric W. Reinheimer[2] & Leonard R. MacGillivray [1,3] ✉

Methods to separate molecules (e.g., petrochemicals) are exceedingly important industrially. A common approach for separations is to crystallize a host molecule that either provides an enforced covalent cavity (intrinsic cavity) or packs inefficiently (extrinsic cavity). Here we report a self-assembled molecule with a shape highly biased to completely enclose space and, thereby, pack efficiently yet hosts and allows for the separation of BTEX hydrocarbons (i.e., benzene, toluene, ethylbenzene, xylenes). The host is held together by N → B bonds and forms a diboron assembly with a shape that conforms to a T-shaped pentomino. A T-pentomino is a polyomino, which is a plane figure that tiles a plane without cavities and holes, and we show the molecule to crystallize into one of six polymorphic structures for T-pentomino tiling. The separations occur at mild conditions while rejecting similarly shaped aromatics such as xylene isomers, thiophene, and styrene. Our observation on the structure and tiling of the molecular T-pentomino allows us to develop a theory on how novel synthetic molecules that mimic the structures and packing of polyominoes can be synthesized and—quite counterintuitively— developed into a system of hosts with cavities used for selective and useful separations.

Separations of petrochemicals are exceedingly important[1]. Processes of utmost importance include separations of benzene, toluene, ethylbenzene, and xylenes, or the BTEX hydrocarbons[1]. BTEX compounds are used extensively in manufacturing and experience tremendous global consumption (i.e., >50 M tons/year). The separation of ethylbenzene from xylenes is troublesome owing to similar reactivities and physical properties (bps: $o$-xylene, 144 °C; $m$-xylene, 139 °C; $p$-xylene, 138 °C, ethylbenzene 136 °C). Rectification processes, for example, are energy intensive, using highly controlled and refined distillations. The removal of ethylbenzene in styrene (bp: 145 °C) production, as well as thiophene (bp: 84 °C) from benzene (bp: 80 °C), also suffer from similar separation issues[2]. Problems of separations are important from economic and sustainability perspectives, representing 10–15% of the world's energy consumption[1]. Indeed, chemical manufacturing greatly benefits from separations of petrochemicals owing to widespread applications in materials and health sciences.

An approach to confront separations of petrochemicals is the crystallization of organic molecules[3–5]. Through molecular recognition, a crystallization can achieve a separation with up to a perfect degree of selectivity (e.g., perfect size exclusion). A molecular host can 'shrink wrap' around a guest to accommodate specific size, shape, and functionality demands of a targeted guest, with the crystallization preferably occurring at ambient temperatures. Given the various challenges that come with developing hosts to recognize, entrap, and sequester molecules, the identification of methods based on molecular and supramolecular designs to facilitate molecular separations represents an important ongoing problem.

Our interest in the molecule **DEPN** lies in its ability to force organic functional groups into close proximity to achieve targeted functions (e.g., catalysis, sensing) (Fig. 1)[6,7]. **DEPN** originates from a family of U-shaped molecules used in molecular recognition, catalysis, and chemical sensing. We expected the closely separated pyridyl groups of **DEPN** to coordinate with up to two B-atoms to afford

[1]Department of Chemistry, University of Iowa, Iowa City, IA 52242, USA. [2]Rigaku Americas Corporation, 9009 New Trails Drive, The Woodlands, TX 77381, USA. [3]Present address: Département de Chimie, Université de Sherbrooke, QC J1K 2R1, Canada. ✉e-mail: leonard.macgillivray2@usherbrooke.ca

the diboron N→B assembly **DBP-DEPN**. Steric considerations suggested that the closely spaced and coordinated catecholate groups would be rotated away from the plane of the naphthyl backbone. Inspired by work of Höpfl, the rigid and conjugated structure of **DBP-DEPN** would exhibit a capacity to allow for inclusion and capture of aromatic-rich hydrocarbons upon crystallization[8,9].

**Fig. 1 | Schemes of chemical structures. DEPN** (left) and **DBP-DEPN** (right).

## Results and discussion

### Molecular T-pentomino structure and packing

**DBP-DEPN** was generated through condensation of **DEPN**, phenylboronic acid, and catechol in dichloromethane. **DBP-DEPN** crystallizes from $CH_2Cl_2$ as thin yellow blades in the monoclinic space $P2_1/c$ (Fig. 2). **DEPN** coordinates with two B-atoms of two catecholates (B→N 1.660(4), 1.664(4)) each of which is canted (117.2(1), 156.7(1)°) and disordered over two positions (occupancies: 88:12 and 85:15). The coordinated pyridyl groups are twisted from coplanarity (117.6(1), 135.7(1)°), with the alkene bonds pointing in opposite directions (Fig. 2a). The separation distance of the pyridyls (N···N 5.998(4)Å) likely reflects steric strain imparted by the closely spaced catecholates. A result of the assembly process is that **DBP-DEPN** adopts an overall T-shaped geometry of edge lengths approximately 10.2 Å × and 12.0 Å and thickness of 8.1 Å (Fig. 2b). C-H•••O hydrogen bonds (C···O: 3.448(4), 3.47(1) Å) reinforce the T-structure (Fig. 2c).

**DBP-DEPN** self-organizes in the solid state within the crystallographic *bc*-plane into face-to-face π-stacks (plane-to-plane: 3.699(6) Å) (Fig. 2d). The pyridyls form head-to-tail dimers with the naphthyls, with the dimers being related by 2-fold rotation within the plane. The

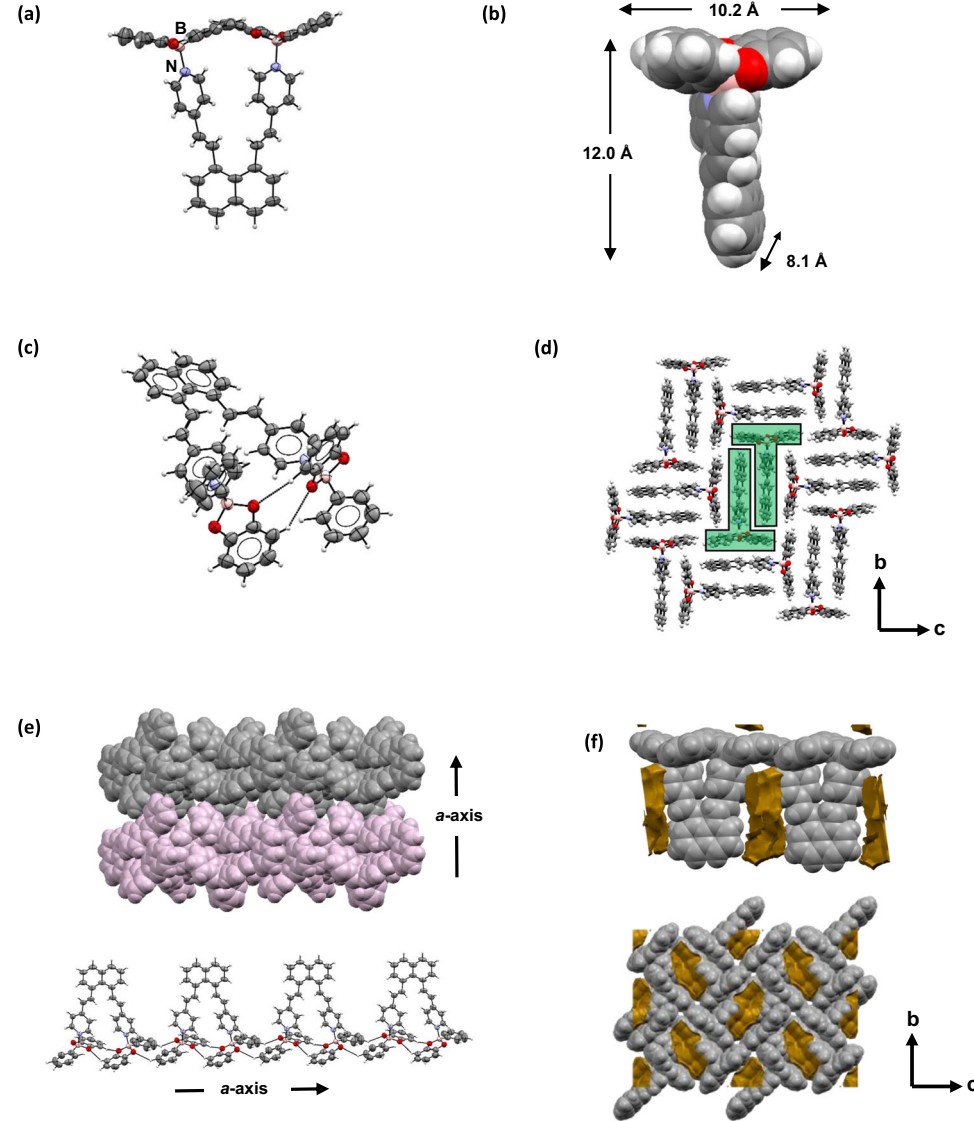

**Fig. 2 | X-ray structure of DBP-DEPN. a** Side view of coordinated assembly, **b** T-shaped geometry, **c** intramolecular C-H•••O hydrogen bonds, **d** 2D packing, **e** tongue-and-groove stacking with intermolecular C-H•••O hydrogen bonds, and **f** voids situated between hosts (orange). Atom colors: O = red, N = blue, B = salmon.

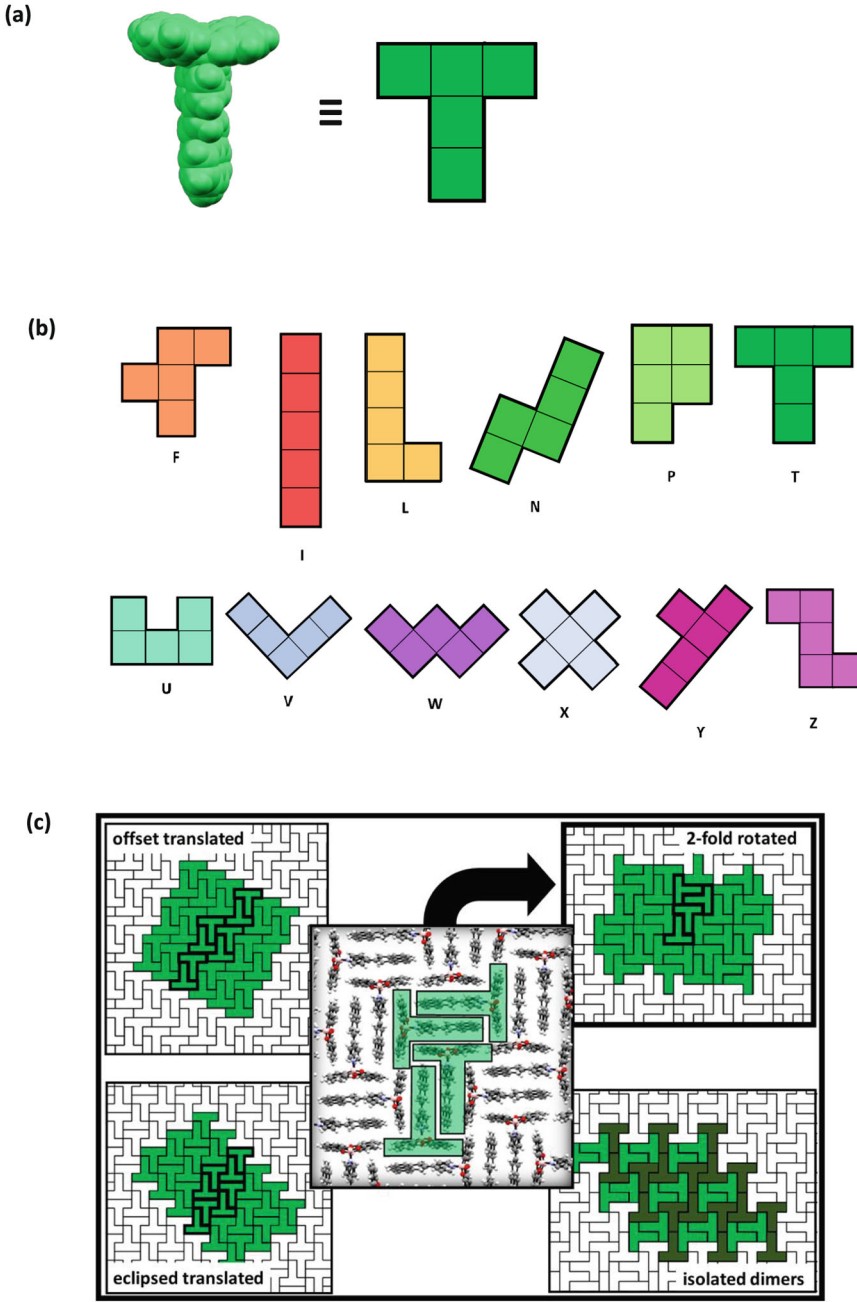

**Fig. 3 | T-pentomino structure and packing. a** T-shape of **DBP-DEPN** and a T-pentomino, **b** the 12 pentominoes, and **c** relation of packing of **DBP-DEPN** to the 2-fold rotated tiling (polymorph). There are four T-pentomino polymorphs that consist of face-to-face stacked dimers. Naming of polymorphs is introduced here.

layers stack along the *a*-axis in a tongue-in-groove fit, with the layers connected by C-H···O hydrogen bonds (3.579(5), 3.74(2) Å) (Fig. 1e). The overall arrangement gives rise to voids filled with $CH_2Cl_2$ solvent molecules (224 Å$^3$) that are situated adjacent to the pyridyl and catecholate groups (Fig. 2f). The void volume corresponds to approximately 5% of the total crystal unit cell volume.

Consultation of plane figure and tiling models reveals the T-shaped molecular structure of **DBP-DEPN** to approximate the shape of T-pentomino (Fig. 3)[10]. A T-pentomino is a plane figure formed by joining five equal squares edge-to-edge in the shape of a T (Fig. 3a). An *n*-omino (*n* = 2, 3, 4, 5...) is a plane figure formed by *n* squares joined at the edges. The popular video game Tetris® is inspired for *n* = 4 and involves attempts to achieve intentional close packing of various tetrominoes. There are 12 possible ways in which the edge-sharing of

squares can generate a pentomino (Fig. 3b). The naming convention of pentominoes corresponds to omino shape in relation to an alphanumeric letter (e.g., L, T, W, X). A feature common of *n*-ominoes is a limited ability of a given *n*-omino to tile a plane by isohedral packing. The resulting tiled structures are packed perfectly, being completely devoid of cavities and holes. Tiling models show the 2D packing of **DBP-DEPN** to conform to one of the six packings of a T-pentomino in a plane (see Supplementary Fig. 18). Four of the six packings, which are polymorphs[11], involve the head-to-tail dimers present in crystalline **DBP-DEPN** (Fig. 3c). The structure reported here conforms to the denoted '2-fold rotated' polymorph. The remaining three polymorphs involve dimers of T-pentominoes related by either translation (i.e., eclipsed translation, offset translation) in 1-isohedral packing or isolated in 2-isohedral packing[12].

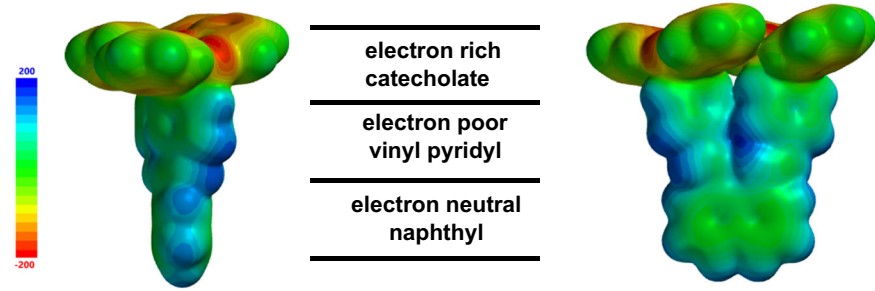

**Fig. 4 | DFT calculations.** Electronically differentiated regions of T-shaped **DBP-DEPN**.

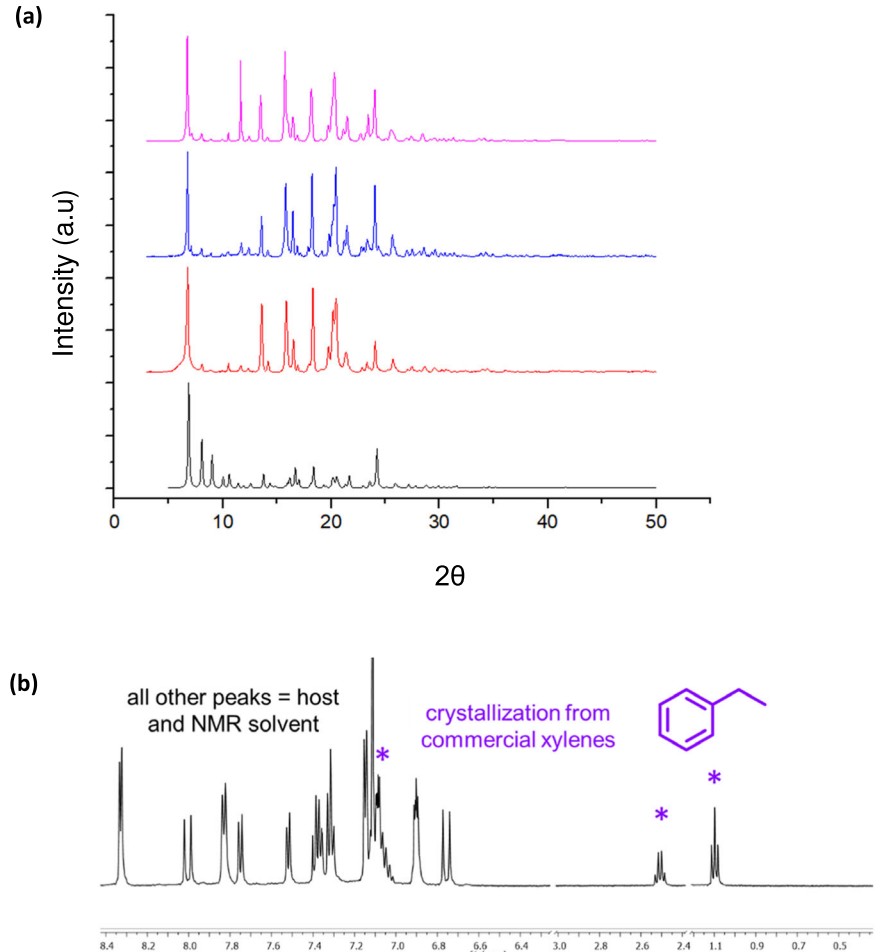

**Fig. 5 | DBP-DEPN solid-state behavior. a** PXRD data of guest lattices and **b** [1]H NMR data demonstrating selective guest uptake of **B3** (asterisks) from xylene mixture.

The 2D packing of T-shaped **DBP-DEPN** can be further understood from electronic considerations. Using DFT calculations, **DBP-DEPN** exhibits three electronically differentiated regions: (i) electron-poor pyridyls, (ii) electron-rich catecholates, and (iii) neutral to slightly electron-rich naphthyls (Fig. 4). Stacking of the electronic-rich catecholates, as encountered in the eclipsed translated and offset translated polymorphs, is generally avoided in the 2-fold rotated tiled structure, whilst head-to-tail pyridyl and naphthyl stacking is achieved. The presence of the voids despite the T-shape of **DBP-DEPN** can be attributed to deviations[13] of the overall shape of **DBP-DEPN** from an ideal T-pentomino. For **DBP-DEPN**, the length of the vertical line (12.0 Å) that intersects with the T's horizontal line (10.4 Å) differs by 1.5 Å (i.e., nearly a carbon-carbon bond length). Nevertheless, the ability of **DBP-DEPN** to tile the plane as a polymorph of a pentomino

attests to robustness of the self-assembly process to achieve the prescribed pattern of close packing.

## Inclusion of BTEX hydrocarbons

The tiling and crystal packing of T-shaped **DBP-DEPN** hosts the BTEX hydrocarbons benzene (**B1**), toluene (**B2**), and ethylbenzene (**B3**) (Fig. 5)[1]. Specifically, crystallization of **DBP-DEPN** from solutions of each of **B1-B3** afforded long yellow needles. SCXRD determinations demonstrated the formation of BTEX-included solids that are isostructural with solvated **DBP-DEPN** (Table 1, Fig. 5a). There are significant changes to the shapes of the cavities as realized by contortions of T-shaped **DBP-DEPN** as reflected by twisting of the pyridyls and catecholates toward 180° versus the solvated structure. The volumes of the cavities shrink up to 17% for **B1** versus the larger guests (Table 2).

**Table 1 | Crystallographic data for DBP-DEPN•guest**

| | CH$_2$Cl$_2$ solvate | Benzene | Toluene | Ethylbenzene |
|---|---|---|---|---|
| CCDC deposition number | 2265493 | 2265492 | 2265491 | 2265490 |
| Empirical formula | C$_{49}$H$_{38}$B$_2$Cl$_2$N$_2$O$_4$ | C$_{54}$H$_{42}$B$_2$N$_2$O$_4$ | C$_{55}$H$_{44}$B$_2$N$_2$O$_4$ | C$_{56}$H$_{46}$B$_2$N$_2$O$_4$ |
| Formula weight/g·mol$^{-1}$ | 811.33 | 797.00 | 818.54 | 832.57 |
| Temperature/K | 150(2) | 100(2) | 100(2) | 100(2) |
| Crystal system | Monoclinic | Monoclinic | Monoclinic | Monoclinic |
| Space group | *P*2$_1$/*c* | *P*2$_1$/*c* | *P*2$_1$/*c* | *P*2$_1$/*c* |
| a/Å | 13.3694(18) | 13.0756(8) | 13.1490(12) | 13.2166(6) |
| b/Å | 21.384(2) | 21.2676(11) | 21.651(2) | 21.8540(8) |
| c/Å | 15.4083(15) | 15.4455(8) | 15.2429(17) | 15.1921(6) |
| α/° | 90 | 90 | 90 | 90 |
| β/° | 103.860(10) | 102.211(4) | 102.590(6) | 102.053(2) |
| γ/° | 90 | 90 | 90 | 90 |
| Volume/Å$^3$ | 4276.8(8) | 4198.0(4) | 4235.2.3(8) | 4291.3(3) |
| Z | 4 | 4 | 4 | 4 |
| $\rho_{calc}$/g·cm$^{-3}$ | 1.260 | 1.261 | 1.282 | 1.289 |
| $\mu$/mm$^{-1}$ | 1.735 | 0.618 | 0.625 | 0.625 |
| F(000) | 1668 | 1670 | 1720 | 1752 |
| Crystal size/mm$^3$ | 0.095 × 0.075 × 0.03 | 0.22 × 0.107 × 0.085 | 0.209 × 0.180 × 0.167 | 0.359 × 0.301 × 0.090 |
| Radiation | CuKα ($\lambda$ = 1.54178) | CuKα ($\lambda$ = 1.54178) | CuKα ($\lambda$ = 1.54178) | CuKα ($\lambda$ = 1.54178) |
| Θ range for data collection/° | 3.606–67.075 | 3.458–67.069 | 3.605–67.068 | 5.286 to 52.744 |
| Index ranges | −15 ≤ h ≤ 15, −25 ≤ k ≤ 25, −17 ≤ l ≤ 18 | −15 ≤ h ≤ 15, −21 ≤ k ≤ 25, −18 ≤ l ≤ 15 | −15 ≤ h ≤ 15, −25 ≤ k ≤ 25, −18 ≤ l ≤ 17 | −15 ≤ h ≤ 15, −25 ≤ k ≤ 26, −16 ≤ l ≤ 18 |
| Reflections collected | 45110 | 51540 | 73408 | 75117 |
| Independent reflections | 76329 [$R_{int}$ = 0.0667, $R_{sigma}$ = 0.0420] | 7504 [$R_{int}$ = 0.0506, $R_{sigma}$ = 0.0284] | 7496 [$R_{int}$ = 0.0630, $R_{sigma}$ = 0.0373] | 7657 [$R_{int}$ = 0.0757, $R_{sigma}$ = 0.0477] |
| Data/restraints/parameters | 76329/355/711 | 7504/559/691 | 7496/613/753 | 7657/552/733 |
| Goodness-of-fit on $F^2$ | 1.0450 | 1.068 | 1.023 | 1.055 |
| Final R indices [$I \geq 2\sigma$ (*I*)] | $R_1$ = 0.0960 | $R_1$ = 0.0781 | $R_1$ = 0.0825 | $R_1$ = 0.0998 |
| | $wR_2$ = 0.2718 | $wR_2$ = 0.2132 | $wR_2$ = 0.2235 | $wR_2$ = 0.2826 |
| R indices (all data) | $R_1$ = 0.1230 | $R_1$ = 0.0895 | $R_1$ = 0.0969 | $R_1$ = 0.1098 |
| | $wR_2$ = 0.3080 | $wR_2$ = 0.2250 | $wR_2$ = 0.2384 | $wR_2$ = 0.2959 |
| Largest diff. peak/hole/e·Å$^{-3}$ | 0.97/−0.71 | 0.88/−0.92 | 0.53/−0.42 | 0.73/−0.60 |

**Table 2 | Geometric parameters describing of T-shape of DBP-DEPN•guest**

| | CH$_2$Cl$_2$ solvate | Ethylbenzene | Toluene | Benzene |
|---|---|---|---|---|
| Catecholate twist (°) | 117.6(1), 156.7(1) | 155.5(3), 151(1) | 122.8(1), 128.3(1) | 155.5(1), 154.0(2) |
| Pyridyl twist (°) | 117.5(1), 135.8(1) | 138.5(2), 145.8(2) | 133.7(8), 146.0(8) | 132.4(1), 124(1) |
| N···N separation (Å) | 5.999(4) | 5.950(5) | 5.968(3) | 5.952(4) |
| Solvent accessible surface (Å$^3$) | 224.63 | 217.31 | 203.83 | 186.27 |

The guest-induced contortions and shrinkage likely reflect domination of attractive interactions between **DBP-DEPN** and the BTEX guests. Overall, the structural changes are consistent with a packed host with cavities that shrink wrap around the hydrocarbons[5].

### Highly selective inclusion

We have discovered **DBP-DEPN** to selectively include **B3**. Crystallization of **DBP-DEPN** from commercial xylenes (Oakwood Scientific) rapidly afforded yellow blade-like crystals. A $^1$H NMR spectrum revealed the solid to exclude each of the *o*-, *m*-, and *p*-xylene isomers. Instead, the spectrum showed the presence of only ethylbenzene in a solid of composition **DBP-DEPN•B3** (Fig. 5b). Commercial xylenes consists of approximately 20% ethylbenzene (**B3**), along with 40%

*m*-xylene (**B4**) and up to 20% each of *o*- and *p*-xylene (**B5-B6**). **B3** is commonly present in xylenes as a product of catalytic reforming of petroleum or by way of isolation of xylenes from coal tar[14]. A $^1$H NMR spectrum of our commercial sample revealed the xylenes to contain 20% of **B3**. The inclusion of **B3** is consistent of **DBP-DEPN** being selective for **B3** uptake from the mixture of xylenes. Crystallizations were also performed in simulated xylene mixtures with decreasing amounts of **B3** (i.e., 15%, 10%, 5%). In each case, crystalline **DBP-DEPN • B3** exclusively formed when the host was crystallized from the preformed xylene mixtures (*m*-, *o*-, *p*-isomers 2:1:1). Up to 80% of the original host material was recovered in the xylene crystallizations with the recoveries occurring in practical time periods on the order of 10 min and at ambient conditions.

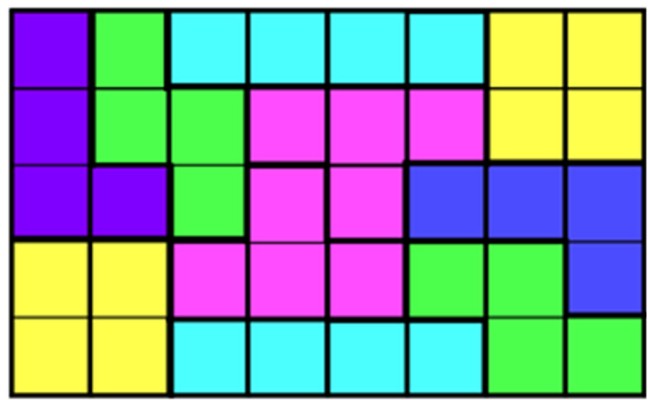

**Fig. 6 | Example of tiling of multiple ominoes.** A rectangle involving a combination of five tetrominoes (not including mirror images for 'L' (blue) and 'Z' (green) combinations).

SCXRD and PXRD data revealed the isolated solid **DBP-DEPN·B3** to be isostructural with the T-pentomino packing arrangement, with **B3** being disordered within the cavities. While the lowest R-refinement value was achieved using a solvent mask, the structures of the aromatic ring and ethyl group of **B3** were clearly apparent. Selective BTEX guest uptake was also realized when **DBP-DEPN** was introduced to either thiophene or styrene[15]. Thiophene is a common impurity in benzene production and styrene is generated from dehydrogenation of **B3**[16]. Crystallizations of **DBP-DEPN** from preformed solutions in competition experiments (1:1 molar ratios) containing either **B1** and thiophene or **B3** and styrene afforded exclusively crystalline **DBP-DEPN·B1** and **DBP-DEPN·B3**. Each solid, thus, excluded thiophene and styrene, respectively. Up to 84% and 89% of the original host was recovered in the **B1**-thiophene and **B3**-styrene crystallizations, respectively. The packing arrangements were isostructural with the T-pentomino lattice. We also note that when competition experiments were performed involving solutions containing two and up to three components of **B1-B3**, a hierarchy of **B3** > **B2** > **B1** was established that is consistent with the high selectively realized involving the xylenes. On the order of 80% of the host was recovered with the solids exhibiting preferential 6:3:1 uptake of **B3:B2:B1**. Variabilities of all cell parameters for the solids characterized by SCXRD were on the order 0.30%, which is consistent with the overall 2D packing not being appreciably pliable.

### Design and polyomino framework theory
Within the field of host-guest chemistry, **DBP-DEPN** acts akin to an extrinsically porous organic molecule[3]. Such molecules are designed to feature unusual molecular and supramolecular structures and conformations that are expected to pack inefficiently[17] so as to *de facto* generate cavities to accommodate guests. The structure of **DBP-DEPN** differs from those hosts in that the T-shape is a priori biased to exhibit packings that lack pores and cavities. In fact, the 2D packing of **DBP-DEPN** agrees with the theory of polyomino tiling that states that *n*-ominoes will exhibit limited packing arrangements that enclose space in two dimensions. Given that the solid **DBP-DEPN** also entraps molecules, it should now, in principle, be possible to synthesize molecules with shapes that conform to the various *n*-ominoes and, in line with the properties of **DBP-DEPN**, develop solids that - counterintuitively - exhibit host-guest behavior. Considerations of perfect 2D packing of organic molecules on surfaces in relation to the presence or absence of voids has been discussed[18]. For the current case, such materials can be expected to create small pores given the biased nature of an omino to enclose space. The field of *n*-ominoes is also replete with examples wherein combinations[19] of different *n*-ominoes pack to fill space (*vis-a-vis*, molecular cocrystals) (Fig. 6). The structures of

*n*-ominoes, effectively, can provide a hitherto undocumented roadmap for the design and discovery of functional organic molecules and host-guest materials.

To conclude, we have reported on the diboron assembly **DBP-DEPN** to exhibit a structure that approximates a T-pentomino. The assembly packs to assume a structure that corresponds to a known polymorph devoid of cavities and holes. The resulting solid selectively entraps BTEX hydrocarbons. We are now exploring the synthesis of molecules with structures that conform to polyominoes, where there will be a focus on how similar a molecule must be to a polyomino for the crystal packing to be realized. It is also intriguing how the single host **DBP-DEPN** separates different guests selectively from different mixtures. The behavior generally contrasts those porous solids that exhibit fixed channels and pores (e.g., zeolites, metal-organic frameworks)[20] and are usually designed for a particular separation. In that way, **DBP-DEPN** is attractive from sustainability and economic perspectives since needs to develop compositionally new host materials for different guests and separations can be obviated.

## Methods
### Chemicals
All materials were obtained from commercial sources and used as received unless indicated. Dibromonaphthalene and palladium bis-triphenylphosphine dichloride were purchased from A2B Chem. Catechol, 4-vinyl pyridine, dichloromethane (99%) toluene (99%), and triethylamine were all purchased from Sigma-Aldrich. Xylenes (95%) and ethylbenzene (99%) were purchased from Acros. Phenylboronic acid was purchased from Oakwood chemicals.

### DEPN
To a 150 ml round bottom flask containing a degassed (argon, 10 min) solution of 50 ml acetonitrile and 75 ml of triethylamine, 1,8-dibromonaphthalene (20 mmol) was added and stirred until dissolved. Palladium bis-triphenylphosphine dichloride (10 mol%, 0.2 mmol) was added while under argon. After filtering through silica gel to remove polymerized material, 4-vinyl pyridine was added to the reaction mixture and stirred at reflux for 72 h. After TLC (ethyl acetate) showed no bromoarene remaining, the reaction mixture was cooled and rotary evaporated until solid remained. This material was eluted through a flash column using ethyl acetate until the solution was clear instead of yellow. The resulting liquid was again rotary evaporated to yield a solid. Recrystallization from ethyl acetate granted shiny, crystalline material after washing with acetonitrile. Yield: 60.3%, White-yellow crystals.

### Assembly formation
**DBP-DEPN.** DEPN (0.9 mmol), phenyboronic acid (1.82 mmol), and catechol (1.82 mmol) are first ground with drops of dichloromethane/toluene (1:1 solution). The resulting powder was scraped from the grinding apparatus and placed into a 30 ml scintillation vial. The mortar-and-pestle were rinsed with another 4.0 ml (2 × 2.0 ml) of solution dropwise and ground to agitate any attached material, which was then added to the vial. The resulting suspension was heated until the powder had dissolved. Upon cooling, translucent bright yellow crystals appeared within 20 min in near quantitative yield. The resulting powder was oven-dried at 80 °C for a minimum of 4 h. Use of dichloromethane is not necessary for crystallization to proceed; however, it allows for ease of solids separation.

**Crystallization experiments.** Aromatic guest-free solid was grown in open air from dichloromethane, resulting in a single molecule inclusion species. For each crystallization with a BTEX hydrocarbon, a scintillation vial containing ground **DBP-DEPN** (0.06–0.08 mmol) was filled with 2.0 ml of BTEX hydrocarbon, gently heated, and allow to cool to room temperature.

Competitive crystallization experiments were performed in the same manner as above but with increased **DBP-DEPN** (0.13–0.16 mmol). Crystallization of **DBP-DEPN** with a 1:1 w/w mixture of thiophene and benzene (2.0 ml) following heating resulted in crystals containing only benzene, as determined by [1]H NMR spectroscopy. PXRD data matched the host material. The procedure was repeated with both a 1:1 w/w mixture of ethylbenzene and styrene (2.0 ml) as well as a standard mixture (3.0 ml) of commercial xylenes (~2(*m*):1(*p*):1(*o*):1(EB)). In addition, competition between guests was undertaken with benzene, toluene, and ethylbenzene being mixed equal parts by mass (0.1 mmol of each, equimolar ratio) with 1:1 ratios for competition between benzene and toluene, benzene and ethylbenzene, toluene and ethylbenzene, and a 1:1:1 ratio of the three together. Crystallizations occurred within 5 min after a 1:20 mass ratio of host to guest was added to the respective solution in a 30 ml scintillation vial and gently heated to ensure dissolution. The crystalline product was removed, and excess solution was allowed to evaporate for 20 min.

**Aromatic guest-free (CH$_2$Cl$_2$ solvate).** [1]H NMR (400 MHz, CDCl$_3$) δ 8.56 (d, $J$ = 5.5 Hz, 4H, H$_f$), 8.15 (d, J = 4, 16 Hz, 2H, H$_h$), 7.99 (d, $J$ = 7.4 Hz, 4H, H$_j$), 7.75–7.73 (m, 4H, H$_c$), 7.72 (d, $J$ = 3.3 Hz, 2H, H$_k$), 7.49 (d, $J$ = 5.5 Hz, 4H, H$_g$), 7.37 (m, $J$ = 3.6 Hz, 4H, H$_d$), 6.98 (d, J = 4, 16 Hz, 2H, H$_h$), 6.88–6.87 (m, 2H, H$_e$), 6.86 (d, $J$ = 2.0 Hz, 4H, H$_b$), 6.75 (d, $J$ = 2.0 Hz, 4H, H$_a$), 5.36 (s, 1H).

Solvate characterization performed using a Exeter Analytical CE-440 addressing carbon, hydrogen, and nitrogen. Analysis calculated for C$_{48}$H$_{36}$N$_2$O$_4$, C: 79.36 H:4.99 N:3.86. Found: C: 73.44 H: 4.77 N:2.83.

**Aromatic guest-free (dried in oven).** [1]H NMR (400 MHz, CDCl$_3$) δ 8.49 (d, $J$ = 5.5 Hz, 4H, H$_f$), 8.16 (d, J = 4, 16 Hz, 2H, H$_h$), 7.89 (dd, 4H, H$_c$), 7.65 (d, 2H, H$_j$), 7.52 (t, 2H, H$_k$), 7.45 (m-overlapped, 4H, H$_d$), 7.43 (m-overlapped, 2H, H$_l$), 7.41 (m-overlapped, 2H, H$_e$), 7.31 (d, $J$ = 5.5 Hz, 4H, H$_g$), 7.17 (m, $J$ = 1.9 Hz, 4H, H$_b$), 6.99 (m, $J$ = 1.9 Hz, 4H, H$_a$), 6.92 (d, J = 4, 16 Hz, 2H, H$_i$).

**Molecular modeling.** Electrostatic potential maps were generated for each **DBP-DEPN** complex by calculations at ground state in gas phase using Spartan '18 V1.2.0 software. Calculations were performed with ωB97X-D/6-31 + G* using the density functional model in a vacuum. Charge value ranges were set to a standard range of +/− 200 KJ/mole with isovalues of 0.002 for each. Single crystal lattice data was used to produce atom coordinates with hydrogens relaxed to neutron distances. Solvent inclusions removed to limit complexity.

**Thermal measurements.** Thermogravimetric analysis data were recorded on a TA Instruments TAQ-500.

## Data availability
The X-ray crystallographic coordinates for structures of **DBP-DEPN**•guest (guest = CH$_2$Cl$_2$, benzene, toluene, ethylbenzene) reported in this study have been deposited at the Cambridge Crystallographic Data Centre (CCDC), under deposition numbers CCDC 2265490 to 2265493, respectively. These data can be obtained free of charge from The Cambridge Crystallographic Data Centre via https://summary.ccdc.cam.ac.uk/structure-summary-form. Details of syntheses, NMR spectral data, and molecular modeling are provided in the Supplementary Material. All relevant data are available from the authors.

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

## Acknowledgements

We thank the National Science Foundation (NSF) for funding (LRM: DMR-2221086, CHE-1828117).

## Author contributions

C.J.H. conceived and performed the syntheses and characterization experiments. E.W.R. performed and finalized the single-crystal X-ray experiments. L.R.M. conceived of the project. All authors contributed to the discussion of the results and writing of the manuscript.

## Competing interests
The authors declare no competing interests.
