## [Peer Review File · Nature Communications]

REVIEWER COMMENTS

Reviewer #1 (Remarks to the Author):

I thank the authors for addressing my previous points. I am generally happy with the responses and would recommend publication in Nature Communications after the following minor points are addressed.

- The CHN analysis of the reported hydrate, while helpful, is not consistent with the proposed formula. I would suggest that hydrate is in fact a DCM solvate ($C_{48}H_{36}B_2N_2O_4 \cdot 1(CH_2Cl_2)$ Calc. C 72.51%, H 4.72%, N 3.45%), which is more plausible given the crystallisation conditions and is also not ruled out by the NMR spectrum. I was able to obtain a reasonable R factor by explicitly refining disordered DCM, although the masking procedure used in the manuscript is certainly appropriate here.

- I am still slightly concerned about the use of raw crystal structure geometries for the DFT calculations. While the qualitative conclusions drawn from the calculations remain valid, best practices are not followed. H atom positions derived from x-ray diffraction data are simply incorrect due to the highly aspherical valence electron density of the H atom. The best approach would be to optimise H atom coordinates (at a modest level of theory) followed by a single point calculation, or alternatively elongating all C-H bonds to neutron distances again followed by a single point calculation.

Reviewer #2 (Remarks to the Author):

In my previous review, I recommended this article be accepted after minor revisions. Upon reading the revised article and author responses to the reviewer comments, I still recommend the article be accepted for publication.

That said, this reviewer still takes slight issue with Figure S10 (previously S6). As plotted, the agreement between simulation and experiment is difficult for a reader to assess. While many peaks do appear to line up, the peak in the experimental data at ~ 16 deg does not appear to agree with the simulation.

Reviewer #3 (Remarks to the Author):

The paper by MacGillivray and co-workers describes how a molecule with a T-shaped can be used to selectively co-crystallise with certain aromatic molecules (e.g. benzene or toluene) as opposed to larger aromatic molecules (e.g. xylenes) and thus function as a method for separation. This is a very simple approach to the isolation and separation of aromatic molecules of industrial importance. It is certainly an innovative idea and therefore has wide potential interest.

I note the previous reviewers comments and note, in particular the comment that suggests that other example(s) would demonstrate the wider applicability of the approach. I had the same feeling when reading the paper and also share the view that it is far from certain that a polyomino strategy would be widely applicable. However, I also agree with the authors that this paper will stimulate this interest and if the community responds as I would anticipate by developing the hypothesis then this approach will become more widely developed in time.

I am sufficiently confident that the field will take up these ideas that I think this preliminary investigation that I am happy to recommend publication of the article in Nature Communications.

My only very brief observation, bearing in mind previous comments from other reviewers, is that in Figure 1(b) it would be better if the molecule had the same orientation as the view in Figure 1(a).

Reviewer #4 (Remarks to the Author):

This manuscript by Hartwick, Reinheimer and MacGillivray reports on the formation of a novel dinuclear boron-nitrogen molecular assembly with T-shaped conformation, capable of exhibiting selective crystallization of molecular species like ethylbenzene present in solvent mixtures such as BTEX hydrocarbons. Perhaps, the most remarkable part of this contribution is the rationalization of these results based on the similarities of the crystal packing of the host molecule DBP-DEPN with 2D tiling of pentominoes. The design of molecules that pack inefficiently in the solid-state generating extrinsically porous materials is very challenging task, and the authors have outlined an interesting proposal for the design and discovery of new host-guest materials related to the geometry and tiling of n-ominoes. Even though, this concept was introduced only with “pentominoes”, it may inspire other researchers to explore further possibilities of multicomponent materials based on these simple geometrical ideas.

The manuscript is clearly presented and well-discussed, the experimental procedures were properly conducted and reliable, particularly, the crystal structure analysis that was challenging given disorder of the guest molecules.

Overall, this work presented an original strategy to target selective sequestration of molecules from industrially important mixtures and will likely attract interest from the broad audience of Nature Communications, especially, in the fields of supramolecular chemistry and crystal engineering.

I would recommend publication after the following points are addressed:

- a) I note that the main host compound DBP-DEPN crystallized as a hydrate, raising the following questions, is water essential for the presence of the voids in the crystal structure? What would be the performance of the anhydrate solid if solvent uptake is done at the solid-vapor interface?
- b) Is it tempting to rationalize the preference of the 2-fold rotated packing solely on electrostatic contributions obtained from MEP by DFT calculations. This qualitative picture worked well for DBP-DEPN, but it may mislead other potential systems given that energetics of the interaction between neutral (yet highly polarized) molecules is governed by dispersion interactions. Therefore, calculations of the interaction energies of the apohost in the crystal lattice may add a significant and informative picture of the real forces influencing 2D-tiling, and this can be done with computational accessible approaches like Crystal Explorer (J. Appl. Crystal. 2021, 54, 1006-1011).

- c) I suggest incorporating numbers on the selectivity observed in the system ethylbenzene-styrene. This is a very important solvent system and preference of ethylbenzene is scarcely reported in the literature (see for instance Chem. Mater. 2022, 34, 197–202). Therefore, what is the yield on the selective crystallization of DBP-DEPN-B3? or how much is left of the original composition of B3 in solution? Likewise, include numbers in the outcome of the three-component mixture B1-B3 (first paragraph p.7).
- d) Indicate in the main text if competition experiments were performed in 1:1 molar or volume ratios.
- e) Figure 1(b) indicates 10.2 Angstroms in one of the edge lengths while the text in p.3 says 10.4 Angstroms.
- f) There is a missing t in “T-penomino” in the first sentence of conclusions p.8

Reviewer #1 (Remarks to the Author):

I thank the authors for addressing my previous points. I am generally happy with the responses and would recommend publication in Nature Communications after the following minor points are addressed.

- The CHN analysis of the reported hydrate, while helpful, is not consistent with the proposed formula. I would suggest that hydrate is in fact a DCM solvate ($C_{48}H_{36}B_2N_2O_4 \cdot 1(CH_2Cl_2)$ Calc. C 72.51%, H 4.72%, N 3.45%), which is more plausible given the crystallisation conditions and is also not ruled out by the NMR spectrum. I was able to obtain a reasonable R factor by explicitly refining disordered DCM, although the masking procedure used in the manuscript is certainly appropriate here.
Found: C: 73.44 H: 4.77 N:2.83

A: We appreciate the comments of the reviewer and agree that the included solvent is CH_2Cl_2 . Our conclusion is based on a reexamination of the 1H NMR data of a freshly prepared sample; moreover, we have also performed a refinement of the single-crystal data and, in the wake of the NMR data, have managed to assign a disordered CH_2Cl_2 that sits across two positions. For the manuscript, we provide both a solvent-squeezed refinement, as well as a comment in the CIF file that indicates a successful refinement of a disordered solvent. We note that the elemental analysis is in line with the assignment of the solvent. Additionally, we now refer to the solid in the running text as a “ CH_2Cl_2 solvate” and agree with the reviewer that the interpretation does not affect the message of the manuscript.

- I am still slightly concerned about the use of raw crystal structure geometries for the DFT calculations. While the qualitative conclusions drawn from the calculations remain valid, best practices are not followed. H atom positions derived from x-ray diffraction data are simply incorrect due to the highly aspherical valence electron density of the H atom. The best approach would be to optimise H atom coordinates (at a modest level of theory) followed by a single point calculation, or alternatively elongating all C-H bonds to neutron distances again followed by a single point calculation.

A: We thank the reviewer for additional comment concerning the calculations. We have now performed the calculation based on an optimization of the H-atom coordinates and a single point calculation. We have updated Figure 3 of the manuscript and the supplementary material accordingly.

Reviewer #2 (Remarks to the Author):

In my previous review, I recommended this article be accepted after minor revisions. Upon reading the revised article and author responses to the reviewer comments, I still recommend the article be accepted for publication.

That said, this reviewer still takes slight issue with Figure S10 (previously S6). As plotted, the agreement between simulation and experiment is difficult for a reader to assess. While many peaks do appear to line up, the peak in the experimental data at ~16 deg does not appear to agree with the simulation.

A: We appreciate the comment of the reviewer. We have concluded that the interpretation of the simulation of the ethylbenzene **B3** X-ray dataset and the provided experimental X-ray data involving the separations involving styrene and xylenes can be difficult to assess. In response, we now provide an experimental PXRD diffractogram of the host **DBP-DEPN** originally crystallized from ethylbenzene (i.e., the solid used to generate the simulated data) (**Figure S13**). We note the better match of the experimental data and conclude that the challenge of the interpretation is likely owing to how the disorder of ethylbenzene **B3** needed to be treated using the single-crystal data.

Reviewer #3 (Remarks to the Author):

The paper by MacGillivray and co-workers describes how a molecule with a T-shaped can be used to selectively co-crystallise with certain aromatic molecules (e.g. benzene or toluene) as opposed to larger aromatic molecules (e.g. xylenes) and thus function as a method for separation. This is a very simple approach to the isolation and separation of aromatic molecules of industrial importance. It is certainly an innovative idea and therefore has wide potential interest.

I note the previous reviewers comments and note, in particular the comment that suggests that other example(s) would demonstrate the wider applicability of the approach. I had the same feeling when reading the paper and also share the view that it is far from certain that a polyomino strategy would be widely applicable. However, I also agree with the authors that this paper will stimulate this interest and if the community responds as I would anticipate by developing the hypothesis then this approach will become more widely developed in time.

I am sufficiently confident that the field will take up these ideas that I think this preliminary investigation that I am happy to recommend publication of the article in Nature Communications.

My only very brief observation, bearing in mind previous comments from other reviewers, is that in Figure 1(b) it would be better if the molecule had the same orientation as the view in Figure 1(a).

A: We thank the reviewer for the comment. The orientations of both Figures match.

Reviewer #4 (Remarks to the Author):

This manuscript by Hartwick, Reinheimer and MacGillivray reports on the formation of a novel dinuclear boron-nitrogen molecular assembly with T-shaped conformation, capable of exhibiting selective crystallization of molecular species like ethylbenzene present in solvent mixtures such as BTEX hydrocarbons. Perhaps, the most remarkable part of this contribution is the rationalization of these results based on the similarities of the crystal packing of the host molecule DBP-DEPN with 2D tiling of pentominoes. The design of molecules that pack inefficiently in the solid-state generating extrinsically porous materials is very challenging task, and the authors have outlined an interesting proposal for the design and discovery of new host-guest materials related to the geometry and tiling of n-ominoes. Even though, this concept was introduced only with “pentominoes”, it may inspire other researchers to explore further possibilities of multicomponent materials based on these simple geometrical ideas.

The manuscript is clearly presented and well-discussed, the experimental procedures were properly conducted and reliable, particularly, the crystal structure analysis that was challenging given disorder of the guest molecules.

Overall, this work presented an original strategy to target selective sequestration of molecules from industrially important mixtures and will likely attract interest from the broad audience of Nature Communications, especially, in the fields of supramolecular chemistry and crystal engineering.

A: We thank the reviewer for the positive comments.

I would recommend publication after the following points are addressed:

a) I note that the main host compound DBP-DEPN crystallized as a hydrate, raising the following questions, is water essential for the presence of the voids in the crystal structure? What would be the performance of the anhydrate solid if solvent uptake is done at the solid-vapor interface ?

A: We note the important comments of the reviewer and generally believe that the suggestion of the reviewer is grounds for more longer-term studies. As described above, we now identify the solid as a CH₂Cl₂ solvate versus hydrate; moreover, it is likely that longer-term work will be needed to identify a solid that would be suitable for solid-vapor interface studies. At the same time, we greatly appreciate the comments of the reviewer in recognizing the importance and need to communicate the *n*-omino design strategy.

b) Is it tempting to rationalize the preference of the 2-fold rotated packing solely on electrostatic contributions obtained from MEP by DFT calculations. This qualitative picture worked well for DBP-DEPN, but it may mislead other potential systems given that energetics of the interaction between neutral (yet highly polarized) molecules is governed by dispersion interactions. Therefore, calculations of the interaction energies of the apohost in the crystal lattice may add a significant and informative picture of the real forces influencing 2D-tiling, and this can be done with computationally accessible approaches like Crystal Explorer (J. Appl. Crystal. 2021, 54, 1006-1011).

A: We greatly appreciate the comments of the reviewer. We are agreeable that the qualitative picture does work well for the 2-fold rotated packing. We also believe that the suggestion of the reviewer is grounds for more longer-term studies. Specifically, to generate a comprehensive detailed view we foresee an opportunity to employ the approaches described by the reviewer for the other polymorphs of the **DBP-DEPN** pentomino system and then later other omino tilings. Moreover, we note that a similar concern was not raised by the other reviewers while, at the same time, there remains an element of urgency to disseminate the effectiveness of the BTEX separations by the current **DBP-DEPN** system.

c) I suggest incorporating numbers on the selectivity observed in the system ethylbenzene-styrene. This is a very important solvent system and preference of ethylbenzene is scarcely reported in the literature (see for instance Chem. Mater. 2022, 34, 197–202).

A: We have added numbers on the selectivity in the system ethylbenzene-styrene. Specifically, we have added that “Up to 84% and 89% of the original host was recovered in the **B1**-thiophene and **B3**-styrene crystallizations, respectively”. We have also added the reference, which is now reference 23.

Therefore, what is the yield on the selective crystallization of DBP-DEPN-B3? or how much is left of the original composition of B3 in solution? Likewise, include numbers in the outcome of the three-component mixture B1-B3 (first paragraph p.7).

A: We thank the reviewer for the suggestions. We now report the yield on the selective crystallization of the **DBP-DEPN-B3** system. Specifically, we indicate in the running text that ‘Up to 80% of the original host material was recovered in the xylene crystallizations’. Additionally, we now include numbers on the outcome of the three-component mixture **B1-B3**. In particular, we indicate that ‘On the order of 80% of the host was recovered with the solids exhibiting preferential 6:3:1 uptake of **B3:B2:B1**’.

d) Indicate in the main text if competition experiments were performed in 1:1 molar or volume ratios.

A: Added. We have indicated that the experiments were performed in 1:1 molar ratios.

e) Figure 1(b) indicates 10.2 Angstroms in one of the edge lengths while the text in p.3 says 10.4 Angstroms.

A: The matter has been addressed with the two numbers now being consistent.

f) There is a missing t in “T-penomino” in the first sentence of conclusions p.8

A: Corrected.

REVIEWERS' COMMENTS

Reviewer #1 (Remarks to the Author):

I thank the authors for addressing my concerns. I am happy to recommend publication with no further comments.

Reviewer #4 (Remarks to the Author):

In this revised version, the authors have addressed sufficiently all reviewers' comments. I think the manuscript is now suitable for publication in Nature Communications

Response to reviewers

It is noted that two reviewers recommended publication and further revisions were not necessary.